# Effect of Growth Mindset on Mental Health Two Years Later: The Role of Smartphone Use

**DOI:** 10.3390/ijerph19063355

**Published:** 2022-03-12

**Authors:** Xiaoxiong Lai, Chang Nie, Shunsen Huang, Yajun Li, Tao Xin, Cai Zhang, Yun Wang

**Affiliations:** 1State Key Laboratory of Cognitive Neuroscience and Learning, Beijing Normal University, Beijing 100875, China; xxlai@mail.bnu.edu.cn (X.L.); huangss@mail.bnu.edu.cn (S.H.); 2Collaborative Innovation Center of Assessment for Basic Education Quality, Beijing Normal University, Beijing 100875, China; niechang@mail.bnu.edu.cn (C.N.); xintao@bnu.edu.cn (T.X.); zhangcai@bnu.edu.cn (C.Z.); 3Guangming Institute of Education Sciences, Shenzhen 518107, China; liyajun0371@163.com

**Keywords:** growth mindset, smartphone use for entertainment, problematic smartphone use, anxiety, depression

## Abstract

The negative association between the growth mindset and mental health problems suggests that prevention and intervention programs to improve mental health by targeting mindset may have potential clinical value. However, research on the longitudinal effect of mindset on adolescent mental health and its underlying mechanisms is lacking. Using a three-wave longitudinal design, we obtained data from a diverse sample of Chinese adolescents (*n* = 2543). Longitudinal multiple mediation models were constructed to examine the effects of the growth mindset on levels of anxiety and depression two years later. In addition, the mediating effects of smartphone use for entertainment and problematic smartphone use (PSU) were examined. After controlling for various covariates and the autoregressive effects of mental health problems, the growth mindset had significant negative effects on anxiety (*β* = −0.053, *p* = 0.004) and depression (*β* = −0.074, *p* < 0.001). Smartphone use had a significant mediating role in the effect of mindset on anxiety (*β* = −0.016, *p* < 0.001) and depression (*β* = −0.016, *p* < 0.001). The growth mindset has long-lasting positive effects on adolescent mental health. Smartphone use for entertainment and PSU mediate the effect of mindset on adolescent mental health.

## 1. Introduction

Adolescence is the time window in which various mental health problems manifest [1]; it is estimated that more than 13% of adolescents, 10–19 years old, have a mental illness as defined by the World Health Organization [2]. Adolescent mental health issues are a global public health problem that impose a heavy burden on the economy worldwide [3]. Given the increasing diagnosis rates and expenses, identifying modifiable prevention and treatment targets that result in significant reductions in symptoms associated with mental health problems is important and can inform large-scale treatment efforts. As an intervention target, enhancement of which can effectively improve academic performance, growth mindset has attracted increasing attention from researchers [4,5,6]. Recently, researchers have looked beyond academic achievement to explore the relationship between mindset and mental health [7,8]. However, except for very few intervention studies [9], existing research on the growth mindset and mental health has typically used a cross-sectional design, which makes it difficult to draw valid inferences about the magnitude and direction of associations and to determine the persistence of the mindset effect. Additionally, there is very little research on the mechanisms of growth mindset effects on mental health, rather than academic achievement; thus, little is known about how the growth mindset affects mental health [7]. Therefore, the present study used a longitudinal design to examine the far-reaching effects of the growth mindset on adolescent mental health. Additionally, the potential mediating role of smartphone use, which is considered a coping strategy related to mental health in modern society [10,11], was investigated.

### 1.1. The Growth Mindset and Mental Health

With the growth mindset, individuals believe that their abilities develop and change over time and that they can be improved through effort; in contrast, the fixed mindset assumes that people’s abilities are innate and difficult to change [12]. Developed from implicit theory, mindset theory emphasizes the importance of human beliefs about their nature and character [12,13]. These beliefs can have a broad and lasting impact on human behavioral tendencies and outcomes, especially when individuals face adversity [8]. Currently, most research on mindset focuses on its relationship with student academic achievement [5]. It is widely accepted that the growth mindset is associated with students’ academic goal orientation, beliefs about effort, and behavioral strategies, which in turn lead to differential academic achievement [14,15]. As research in this area has expanded, the impact of the growth mindset on self-regulation and its role in various outcomes has attracted increasing attention [8,16]. When dealing with adverse or challenging situations (not limited to the academic sphere), people’s beliefs about their abilities may lead to different outcomes by influencing their behavioral intentions and coping strategies [7,8].

Mental health is important to the quality of life and lifelong development [2]. Adolescents’ maladaptive perceptions, such as beliefs that their current condition will not change and nonadaptive responses to negative life events, predict mental health problems [17,18]. Furthermore, many evidence-based psychotherapies, such as cognitive behavioral therapy, have demonstrated that patients’ psychopathological symptoms can be altered by changing their feelings, thoughts, and behaviors [19]. Thus, there is reason to believe that a growth mindset is a potential protective factor against mental health problems. Research extending mindset effects to mental health contexts has promising early findings [7,20]. For example, a meta-analysis found a moderate negative correlation (*r* = −0.220, 95% CI = −0.257, −0.184, *p* < 0.001) between the growth mindset and mental health problems (i.e., anxiety, depression, psychological stress, and lack of well-being) [7]. Large-scale international student assessments have also found that students with a growth mindset experienced increased positive emotions, and this effect was more pronounced in East Asian countries, namely, China, Japan, and South Korea [21]. However, previous research is primarily based on cross-sectional research designs [7,20]. Exploring the longitudinal relationship between the growth mindset and mental health is important to understand the causal relationship between the two and to evaluate the long-term protective effect of the growth mindset on mental health. This will provide important insight into the feasibility of mindset-specific prevention and intervention programs for mental health problems. Thus, this study used participant data over a three-year span to examine the longitudinal association between the growth mindset and mental health problems.

### 1.2. Smartphone Use as a Coping Strategy

Accumulating evidence in the field of educational psychology has illuminated the mechanisms underlying the role of mindset on academic achievement, but little is known in the field of clinical psychology [15]. Similar to the mechanisms by which mindset influences academic achievement, researchers have proposed that intentions toward health promoting efforts (e.g., attitudes toward treatment) and choice of coping strategy may be mediating factors in the relationship between mindset and mental health [7,22]. Empirical studies have found significant positive associations between the growth mindset and positive coping strategies and perceptions of treatment value [7]. In recent years, the development of information and communication technology (ICT), especially the rapid spread of smartphones, has had a significant impact on the daily lives of adolescents [23]. Researchers have noted that an increasing number of people are using digital technologies to shape their emotional state, a phenomenon termed digital emotion regulation [11]. Additionally, theoretical models in the field of cyberpsychology, such as the compensatory internet use model [24], self-escape theory [25], and the needs-affordances-features model of technology use [26], point out that people may use ICT (e.g., smartphones) to escape from or resolve negative events or situations, triggering diverse outcomes. Thus, integrating the concept of digital emotion regulation into mindset theory may provide new perspectives to explain the mechanisms by which growth mindset affects mental health in contemporary society.

Of the various smartphone uses, the role of smartphone use for entertainment on mental health is controversial [10]. Unlike some other smartphone use behaviors that contribute to problem solving (e.g., using a smartphone to ask for help or gather useful information) [27,28], using a smartphone for entertainment, while distracting from a negative event or situation in the short term [29], may not be beneficial to mental health in the long term [30]. In various theoretical models, the use of ICT for entertainment has been identified as an antecedent of problematic technology use (e.g., problematic smartphone use, PSU) [31,32,33]. PSU may disrupt an individual’s social connections and lead to sleep disruption, thereby exacerbating mental health problems [34,35,36,37]. In contrast, a growth-oriented mindset in modern society may benefit mental health by reducing nonadaptive coping strategies (e.g., the use smartphones for entertainment). However, few studies have examined the mechanisms of mindset effects on mental health, and even fewer studies have utilized a longitudinal design. Therefore, this study used a longitudinal design to examine the role of smartphone use in the relationship between mindset and mental health.

### 1.3. The Present Study

This study had two specific aims. First, we examined the longitudinal effects of the growth mindset on mental health. We controlled for the effects of sex, age, and SES, as well as the autoregressive effects of mental health problems. Anxiety and depression, as proxy variables for mental health problems, were separately assessed as outcome variables, thus testing whether the mindset effect was stable across different kinds of mental health problems. Second, we examined the influence of smartphone use for entertainment and PSU on the relationship between mindset and mental health. As mentioned earlier, adolescence is a window for the onset of various psychological disorders. This study focused on the far-reaching effects of growth mindset on adolescent mental health and therefore selected a sample of students who were approaching or just entering adolescence (grades 3–7).

## 2. Materials and Methods

### 2.1. Participants

In this longitudinal study, students in grades 3 to 7 from four cities in central China were surveyed. Participants were investigated at three time points (April 2019, July 2020, and April 2021). A total of 2543 students (average age = 11.54 ± 1.84 years, 1245 females) participated in this longitudinal study. Of these, 2543, 2090, and 2277 students participated in Waves 1, 2, and 3 of the study, respectively. Data for Waves 1 and 3 were collected in school, while data for Wave 2 were collected online. The procedure for Waves 1 and 3 involved students independently filling out questionnaires, with one parent of each participant reporting information related to the family environment, such as parental education and annual household income. In Wave 2, due to the COVID-19 pandemic, a link to an online questionnaire was sent to participants via WeChat, a popular communication application that is easily accessible by students in China. This study was approved by the Institutional Review Board (IRB) of State Key Laboratory of Cognitive Neuroscience and Learning, Beijing Normal University. Informed consent was obtained from guardians, teachers, and school administrators for this study. The demographic characteristics of participants is shown in Table 1.

### 2.2. Measures

Growth mindset. The growth mindset inventory developed by Dweck for adolescents was used to measure participants’ growth mindset in Wave 1 [12]. We used a Chinese version of this inventory [38], which consists of four items (e.g., “No matter who you are, you always can change your intelligence a lot”). Students were asked to rate their agreement with these statements on a 6-point Likert scale ranging from 1 = strongly disagree to 6 = strongly agree. Higher scores indicated higher levels of growth mindset. In this study, Cronbach’s α was 0.74.

Smartphone use for entertainment. A mobile phone-use patterns questionnaire developed by Jiang and Zhao for young Chinese people was adapted to measure students’ use of smartphones for entertainment in Wave 2 [39,40]. We used the entertainment dimension of this questionnaire, which includes five items assessing game playing, watching funny videos, listening to music, and so on. We asked students to rate their frequency of the described smartphone uses on a 4-point Likert scale ranging from 1 = never to 4 = very frequently. Higher scores indicated more frequent use of smartphones for entertainment. The Cronbach’s α in this study was 0.83.

Problematic smartphone use. The smartphone addiction proneness scale for adolescents developed by Kim, Lee, Lee, Nam, and Chung was used to measure PSU in Wave 2 [41]. We used a Chinese revision of this scale [42], which consists of 16 items and includes four dimensions: (1) disturbance of adaptive functions; (2) withdrawal; (3) tolerance; and (4) virtual life orientation. Each dimension was assessed with four items (e.g., “It would be painful if I am not allowed to use a smartphone”). Students were asked to rate the frequency of the described events on a 4-point Likert scale ranging from 1 = never to 4 = always. Higher scores indicated a higher propensity for PSU. In this study, Cronbach’s α was 0.89.

Anxiety. The Generalized Anxiety Disorder (GAD-7) scale was used to measure anxiety in Waves 2 and 3 [43]. The appropriateness of this scale has been demonstrated in a sample of Chinese adolescents [44]. It consists of seven items (e.g., “I was not able to stop or control worrying”). Students were instructed to “Think about the last 2 weeks. How often were you been bothered by the following problems?”. They were then asked to rate the frequency of the described problems a 4-point Likert scale ranging from 1 = not at all to 4 = nearly every day. Higher scores indicated higher levels of anxiety. The Cronbach’s α in this study was 0.96.

Depression. The Center for Epidemiologic Studies Depression (CESD-10) scale was used to measure depression in Waves 2 and 3 [45]. The appropriateness of this scale in the Chinese adolescents has been confirmed [46]. It consists of ten items (e.g., “I felt everything was an effort”). Students were instructed to “Think about how often each event occurred during the past week”. They were then asked to rate the event frequency on a 4-point Likert scale ranging from 1 = rarely or none of the time to 4 = most or all the time. Higher scores indicated higher levels of depression. The Cronbach’s α in this study was 0.84.

Covariates. The demographic covariates of sex (1 = male, 2 = female), age, and family socioeconomic status (SES) were controlled for in the models. Based on the methods of previous studies [47], the standard scores of the participants’ paternal education level, maternal education level, and annual household income in 2018 were averaged to synthesize an SES index, with higher scores indicating higher SES.

### 2.3. Data Analysis

We conducted descriptive statistics and missing value analysis in SPSS (version 25, IBM: Armonk, NY, USA) and the SPSS Missing Value Analysis module. In this study, the average missing rates for each variable were 4.01%, 0.71%, and 1.18% in Waves 1, 2, and 3, respectively. We used a maximum likelihood (ML) estimation procedure to estimate missing item-level data and an expectation-maximization imputation method for missing case-level data [48]. We conducted the multiple mediation models in Mplus 7.4 [49]. Sex, age, and SES were included as covariates in both models. Chi-square values, the comparative fit index (CFI), Tucker–Lewis index (TLI), and root mean square error of approximation (RMSEA) were used to evaluate the overall model fit. In addition, to test the significance of the mediating effects, bias-corrected bootstrap tests were performed with a 95% confidence interval (CI) [50].

## 3. Results

### 3.1. Descriptive Statistics

The means, standard deviations, and correlations for growth mindset, smartphone use for entertainment, PSU, anxiety, and depression are shown in Table 2. There were significant correlations between all study variables.

### 3.2. Effect of the Growth Mindset on Subsequent Anxiety

We included smartphone use for entertainment, PSU, and anxiety in Wave 2 between the growth mindset assessed in Wave 1 and anxiety assessed in Wave 3 to build a multiple mediation model (see Figure 1). This model showed good fit indices [*χ*^2^(1) = 0.017, *p* = 0.898; CFI = 1.00; TLI = 1.00; RMSEA (90% CI) = 0.00 (0.00, 0.02)]. As shown in Table 3, both the direct and total indirect effects were significant according to the bias-corrected bootstrap approach, indicating that the growth mindset directly predicted anxiety two years later, as well as smartphone use at this time. The indirect effects accounted for 30.19% of the total effect. The model results suggest that the growth mindset negatively predicted anxiety up to two years later, even after controlling for various covariates (i.e., sex, age, and SES) and the autoregressive effect of anxiety in Wave 2.

### 3.3. Effect of the Growth Mindset on Subsequent Depression

We included smartphone use for entertainment, PSU, and depression in Wave 2 between the growth mindset assessed in Wave 1 and depression assessed in Wave 3 to construct a multiple mediation model (see Figure 2). This model showed good fit indices [*χ*^2^(1) = 7.192, *p* = 0.007; CFI = 1.00; TLI = 0.95; RMSEA (90% CI) = 0.05 (0.02, 0.09)]. As shown in Table 3, both the direct effect and the total indirect effects tested by the bias-corrected bootstrap approach were significant, indicating that the growth mindset directly predicted depression two years later as well as smartphone use at this time. The indirect effects accounted for 21.62% of the total effect. The model results indicated that the growth mindset still negatively predicted depression up to two years later, even after controlling for various covariates (i.e., sex, age, and SES) and the autoregressive effect of depression in Wave 2.

## 4. Discussion

This study found that the growth mindset was a negative predictor of mental health problems two years later, even after controlling for various covariates and the autoregressive effects of mental health problems. Smartphone use significantly mediated the relationship between mindset and mental health, indicating that smartphone use for entertainment and PSU are nonnegligible coping strategies in a digital society that may affect adolescents’ mental health. The two common mental health problems measured, anxiety and depression, were consistent in each model, demonstrating the stability of the mindset effect.

### 4.1. Effect of the Growth Mindset on Subsequent Mental Health

The results of this study suggest that the growth mindset has long-lasting positive effects on mental health. This finding complements those of previous research and suggests the potential clinical value of mindset-specific prevention and intervention programs for mental health. Although the effect size of the growth mindset on mental health problems in this study was not large, researchers suggest that the criteria of cross-sectional studies should not be used to interpret effect sizes when controlling for stability effects in longitudinal autoregressive models [51]. In longitudinal studies, even relatively small effect sizes (e.g., *r* < 0.1) may have significant impacts due to accumulation over time [51]. As the effects of a growth mindset in this study spanned a two-year time interval, which is longer than the longest time interval of previous studies (18 months) [52], the results of this study are important. Nevertheless, previous studies have found that developmental characteristics can moderate the effectiveness of mindset interventions on mental health [53], so the findings of the current study should be treated with caution.

### 4.2. The Mediating Effects of Smartphone Use

As one of the few studies to explore the mechanisms by which mindset influences mental health, this study found a significant mediating effect of smartphone use on this relationship. These results suggest that a growth mindset can impact adolescents’ mental health by influencing their coping strategies. The possibility that smartphone use may be a coping strategy for negative events and situations highlights theoretical perspectives on digital emotion regulation and provides supporting evidence for a compensatory internet use model [11,24]. Additionally, this study focused on the specific type of smartphone use, highlighting the importance of differentiating smartphone use into different contexts rather than treating it as uniform (e.g., screen time) to examine its impact on adolescent PSU and other outcomes [10,54,55]. By improving adolescents’ growth mindset, their willingness to use problem-solving apps in smartphones can be increased [27], while their recreational use of smartphones to escape stress can be reduced. Based on previous research and the findings of this study, the combination of growth mindset intervention and smartphone use guidance is a possible avenue for future interventions targeting adolescent mental health, especially in regions such as China, where adolescent smartphone penetration is high and problematic smartphone use is more common [56,57]. The role of PSU in this study suggests that PSU development may be a key link between daily smartphone use and negative outcomes, such as mental health problems [32,33]. In this study, the indirect effects of smartphone use for entertainment and PSU did not explain a high proportion of the total effect, suggesting that other variables may mediate the effect of mindset on mental health, such as psychological resilience [58]. In short, as society becomes more information-based and as the proportion of digital natives in the global population increases [23], the role of ICT use on individual mental health should be studied in greater depth, and traditional theoretical models, such as mindset theory and coping theory [12,59], should be expanded and revised in this context.

### 4.3. Limitations and Future Directions

Although this study presents several important findings regarding the effect of mindset on adolescent mental health, it has several limitations. First, this study was conducted in China; thus, our findings cannot be directly generalized to other countries. As cultural differences are frequently mentioned as factors influencing the mindset effect [15,21], future research should include cross-cultural samples. Second, the adolescents in this study were mainly aged 11–15 years old when they participated in Wave 3, and few participants were in late adolescence. Considering that developmental characteristics are potential factors influencing the mindset effect on mental health, future researchers should expand the age range of participants or extend the follow-up period of longitudinal studies. Third, we used a single method (survey) to collect data from two different sources (adolescents and their parents), the results of which may be affected by social desirability and common method bias. In future studies, a multi-informative data collection method is needed to verify self-reports, such as field observations. Fourth, this study did not measure the research variables in all waves, which made some of the more rigorous statistical tests (e.g., cross-lagged panel model) impossible to implement. We must be cautious about drawing causal conclusions from the results of our study. To determine a strict causal relationship between mindset and mental health, rigorous randomized trials and long-term follow-up cohort studies are needed. Fifth, the outbreak of the COVID-19 pandemic occurred during the time period in which this study was conducted. The pandemic has had an impact on adolescent mental health, which may have affected the results of this study. Future studies should seek to control for the effects of the COVID-19 pandemic or test the findings of this study in the post-pandemic era.

## 5. Conclusions

The growth mindset has a long-lasting positive effect on adolescent mental health. Smartphone use for entertainment and PSU mediate the effect of mindset on adolescent mental health. Prevention and intervention programs for mental health problems targeting mindset may have potential clinical value.

## Figures and Tables

**Figure 1 ijerph-19-03355-f001:**
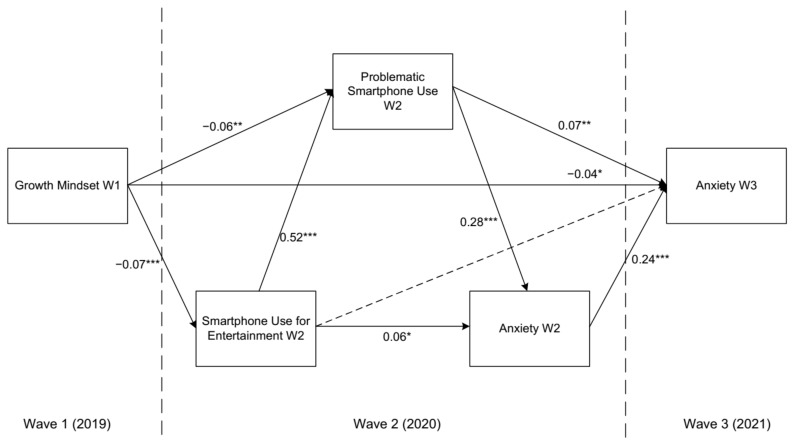
The longitudinal multiple mediation model of the effect of the growth mindset on adolescent anxiety. Statistics are standardized path coefficients. Dotted lines represent a nonsignificant relation. The model also includes sex, age, and SES as covariates in Waves 2 and 3, which are not shown for clarity of presentation. * *p* < 0.05, ** *p* < 0.01, *** *p* < 0.001.

**Figure 2 ijerph-19-03355-f002:**
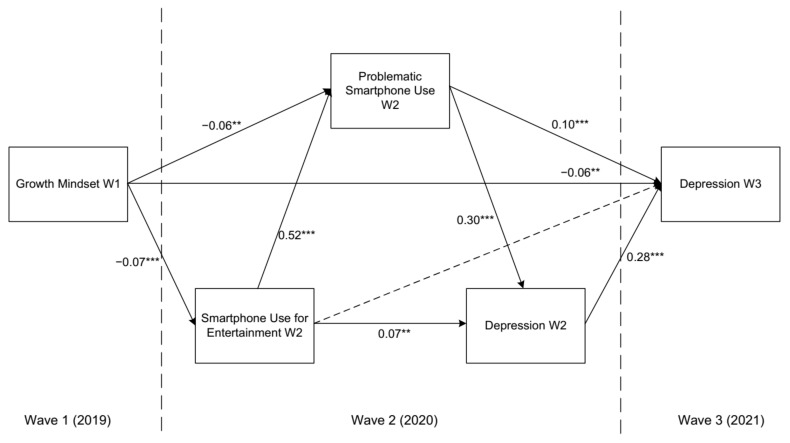
The longitudinal multiple mediation model of the effect of the growth mindset on adolescent depression. Statistics are standardized path coefficients. Dotted lines represent a nonsignificant relation. The model also includes sex, age, and SES as covariates on variables in Waves 2 and 3, which are not shown for clarity of presentation. ** *p* < 0.01, *** *p* < 0.001.

**Table 1 ijerph-19-03355-t001:** Demographic characteristics of participants.

Variable	Group	Percentage
Residence	City	41.2%
	Township	19.7%
	Rural	39.0%
Only child	Yes	92.7%
	No	7.3%
Maternal education	<College	78.1%
	≥College	21.9%
Paternal education	<College	75.2%
	≥College	24.8%
2018 Annual income	<¥50,000	60.3%
	¥50,000–100,000	21.1%
	>¥100,000	18.6%

Note: ¥ = RMB.

**Table 2 ijerph-19-03355-t002:** Descriptive statistics and correlations among the research variables.

	1	2	3	4	5	6	7
1. GM T1							
2. ENT T2	−0.09 ***						
3. PSU T2	−0.10 ***	0.55 ***					
4. Anxiety T2	−0.05 *	0.30 ***	0.37 ***				
5. Anxiety T3	−0.07 ***	0.20 ***	0.21 ***	0.31 ***			
6. Depression T2	−0.11 ***	0.33 ***	0.40 ***	0.78 ***	0.32 ***		
7. Depression T3	−0.11 ***	0.16 ***	0.23 ***	0.29 ***	0.72 ***	0.35 ***	
M	4.17	2.06	1.92	1.44	1.73	1.73	1.80
SD	1.07	0.60	0.63	0.58	0.66	0.49	0.50

Note: GM = growth mindset; ENT = smartphone use for entertainment; PSU = problematic smartphone use; T1 = Time 1; T2 = Time 2; T3 = Time 3. * *p* < 0.05. *** *p* < 0.001.

**Table 3 ijerph-19-03355-t003:** Bias-corrected bootstrap test results on the mediating effects.

Path	*β*	SE	*t*	*p*	95% CI
Model 1 (Anxiety)					
GM1→ENT2→ANX3	−0.003	0.002	−1.709	0.087	[−0.008, 0.000]
GM1→PSU2→ANX3	−0.004	0.002	−2.023	0.043	[−0.009, −0.001]
GM1→ENT2→PSU2→ANX3	−0.002	0.001	−2.077	0.038	[−0.005, −0.001]
GM1→ENT2→ANX2→ANX3	−0.001	0.001	−1.786	0.074	[−0.002, 0.000]
GM1→PSU2→ANX2→ANX3	−0.004	0.001	−2.846	0.004	[−0.007, −0.001]
GM1→ENT2→PSU2→ANX2→ANX3	−0.002	0.001	−3.234	0.001	[−0.004, −0.001]
Total indirect effect	−0.016	0.004	−3.965	<0.001	[−0.026, −0.009]
Direct effect	−0.036	0.018	−1.980	0.048	[−0.074, −0.001]
Total effect	−0.053	0.019	−2.851	0.004	[−0.093, −0.020]
Model 2 (Depression)					
GM1→ENT2→DEP3	0.002	0.002	1.289	0.198	[0.000, 0.007]
GM1→PSU2→DEP3	−0.006	0.002	−2.389	0.017	[−0.012, −0.002]
GM1→ENT2→PSU2→DEP3	−0.004	0.001	−2.524	0.012	[−0.007, −0.001]
GM1→ENT2→DEP2→DEP3	−0.001	0.001	−2.004	0.045	[−0.003, 0.000]
GM1→PSU2→DEP2→DEP3	−0.005	0.002	−2.839	0.005	[−0.008, −0.002]
GM1→ENT2→PSU2→DEP2→DEP3	−0.003	0.001	−3.325	0.001	[−0.005, −0.001]
Total indirect effect	−0.016	0.004	−3.521	<0.001	[−0.026, −0.008]
Direct effect	−0.058	0.019	−3.095	0.002	[−0.099, −0.022]
Total effect	−0.074	0.019	−3.929	<0.001	[−0.112, −0.036]

Note: Number of bootstrap samples = 1000. CI = confidence interval; GM = growth mindset; ENT = smartphone use for entertainment; PSU = problematic smartphone use; ANX = anxiety; DEP = depression. The number after the variable name indicates the wave in which the variable was measured.

## Data Availability

The datasets generated during and/or analyzed during the current study are available from the corresponding author on reasonable request.

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
