# Peer review of "Effect of Growth Mindset on Mental Health Two Years Later: The Role of Smartphone Use"

_ijerph, 2022, doi:10.3390/ijerph19063355_

Round 1

Reviewer 1 Report

Thank you for inviting me to review this paper, which is an innovative and interesting study. However, I have some suggestions for the authors that might help strengthen their manuscript.

Introduction:

The article would be improved if it presented a more in-depth theoretical framework.

Materials and Methods

Participants

Commas to mark quantities in thousands can be confused with decimals, perhaps this problem would be eliminated by leaving the quantities without commas. For example: "A total of 2543 students..." instead of "A total of 2,543 students...".

Could you please justify why a sample of students from grades 3 to 7 was chosen and not from other age groups?

Measures

Some measurement instruments are intended for adolescents, but the sample of participants belongs to an age range closer to childhood than to adolescence. Could you explain if a correct adaptation of the measures to the age range was made?

Smartphone use for entertainment: this heading does not specify in which wave the measure was applied.

It is necessary to further clarify the reasons why each measure is carried out in wave 1, 2 or 3. For example, could you explain why anxiety and depression were not also measured in wave 1? Was it not considered necessary to have a baseline measure with which to compare successive measurements?

Results

In a time as stressful as the Covid-19 pandemic, one would expect some effect on depressive and anxious symptomatology independent of the growth mindset effect. The measurements coincide precisely with the arrival of the pandemic and the period of adaptation to this situation. How then were possible extraneous variables that might influence the observed changes in anxiety and depression over time controlled for? It would be desirable to better justify this issue.

Discussion

If possible, the results should be developed, providing further empirical evidence congruent with the results, and contrasting them with other research that finds different results.

Conclusions

Given that the conclusions heading is so scarce, I recommend that it be merged into a single discussion and conclusions heading.

Reviewer 2 Report

This three-wave longitudinal study examined the association of growth mindset at baseline with anxiety and depression two years later and the mediating effects of smartphone use for entertainment and problematic smartphone use.

This study had several strengths, including large same size, longitudinal study design, and examining the mediating effect of smartphone use. The manuscript was well-organized and well-written. The statistical analysis was reasonable. The results of this study can provide new knowledge to the field of grow mindset study.

I have only one suggestion for the authors to improve the manuscript. Although how to improve grow mindset in children and adolescents is not the major goal of this study, the readers may be interested in the practical intervention for improving grow mindset. The authors may provide some summaries of results of previous studies and propose the possible ways to conduct the intervention in China, especially in non-urban areas.

Round 2

Reviewer 1 Report

Congratulations, the article has improved, although there are some limitations that call for caution with the extrapolation of the results.